# Deformation Mechanisms of FCC-Structured Metallic Nanocrystal with Incoherent Twin Boundary

Yang Tao [1],*, Yufeng Zhao [2], Zhanxin Wang [2], Libo Fu [2] and Lihua Wang [2],*

1   Department of automobile, Huanggang Polytechnic College, Huanggang 438002, China
2   Beijing Key Lab of Microstructure and Properties of Advanced Materials, Beijing University of Technology, Beijing 100124, China; yfz@emails.bjut.edu.cn (Y.Z.); wzx824914309@163.com (Z.W.); fulibo57@126.com (L.F.)
*   Correspondence: taoyang@hgpu.edu.cn (Y.T.); wlh@bjut.edu.cn (L.W.)

**Abstract:** Incoherent twin boundaries (ITBs) can significantly affect the mechanical properties of twin-structured metals. However, most previous studies have focused on the deformation mechanism of the coherent twin boundary (CTB), and metals with ITB-accommodated plasticity still require further investigation. In this study, deformation mechanisms of FCC-structured nanocrystal metals with ITBs were investigated using molecular dynamic (MD) simulations. We revealed that three deformation mechanisms occur in metals with ITBs. The first type of deformation was observed in Au, where the plasticity is governed by partial dislocation intersections with CTBs or reactions with each other to form Lomer–Cottrell (L–C) locks. In the second type, found in Al, the deformation is governed by reversible ITB migration. The third type of deformation, in Ni and Cu, is governed by partial dislocations emitted from the ITB or the tips of the stacking faults (SFs). The observed L–C lock formation, as well as the reversible ITB migration and partial dislocation emission from the tips of SFs, have rarely been reported before.

**Keywords:** incoherent twin boundary; plastic deformation; dislocation; metallic nanocrystal; face-centered cubic

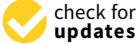



## 1. Introduction

Previous studies have shown that face-centered cubic (FCC) structured metals with growth nanotwins always exhibit ultrahigh strength and excellent ductility [1–5]. As the mechanical properties of materials are directly related to their deformation mechanisms [6–14], revealing the atomic-scale deformation mechanisms of materials is important for understanding their mechanical performance and realizing their desired mechanical properties [11,12,15–18]. In recent decades, a large number of studies have been conducted on the deformation mechanisms of twin-structured metals [16–28]. Many molecular dynamics (MD) simulations and experimental investigations have suggested that the ultrahigh strength of twin-structured metals results from partial dislocations intersecting with coherent twin boundaries (CTBs) [10,20–22]. These studies on the deformation mechanism of CTBs suggest that the strengthening effect in twin-structured metals resulted from dislocation–CTB interactions [24–28]. However, in physical twin-structured metals, there is also a high density of incoherent twin boundaries (ITBs), which can significantly affect their mechanical properties [22,25,29–33]. Thus, revealing the deformation mechanisms of metals with ITBs is also very important. However, most previous studies have focused on the mechanisms of the dislocation–CTB interaction [24,27,34–36], and the effect of the dislocation–ITB interaction on the mechanical properties of metals with ITBs still needs further investigation. In FCC-structured metals, the ITB consists of Shockley partial dislocations on each {111} plane [30–32]. These partial dislocations can move away from the ITB, leading to a decrease in the thickness of the twin or ITB migration [6,18,31,32,37]. However, most studies have focused on Cu [28–31,37], and whether the proposed deformation model can be valid for other metals is unclear. Thus, it will be interesting to



provide systematic evidence on how ITBs accommodate the plastic deformation of various FCC-structured metals.

To address the above-mentioned issues, the deformation mechanisms of metallic Au, Al, Ni, and Cu nanocrystals with ITB were investigated using MD simulations. We showed that three deformation mechanisms, including the partial dislocation intersection with CTBs or reaction with each other to form Lomer–Cottrell (L–C) locks, reversible ITB migration, and partial dislocations emitted from the ITB or the tips of the SFs, occur in different FCC metals.

## 2. Simulation Method

We adopted a rectangular model with periodic boundary conditions along the x-axis. The model was divided into a two-part computational cell, which was formed from the combination of two "L"-shaped parts in order to construct a twin-structured nanocrystal, as shown in Figure 1. Then, this model was relaxed, ensuring a structure with minimum energy. The nanocrystal contained a Σ3 {112} ITB with the thickness of 30 (111) planes, and two CTBs separated by the ITB, as shown in Figure 1. The deformations of twin-structured Au, Al, Ni, and Cu nanocrystals were conducted using MD simulations at room temperature, ~298 K. These twin-structured nanocrystals with a square cross-section contained about 24,000 atoms. The coordinate systems were the x-axis along [112], the y-axis along [11$\bar{1}$], and the z-axis along [1$\bar{1}$0]. The dimensions of the nanocrystals were 12 nm for both the x- and z-axes, and 18 nm for the y-axis. In the three-dimensional structure of the sample, as shown in Figure 1a, the ITB (gray atoms) was parallel to the longitudinal direction of the nanocrystals, and the CTBs (red atoms) were perpendicular to that direction. Stretch simulations were conducted by using a large-scale atomic/molecular massively parallel simulator (LAMMPS) program [38], with the classical embedded-atom method (EAM) potential [39–42]. The ITB consisted of periodical sets of partial dislocations " $b_1$, $b_2$, $b_3$" with $b = \frac{1}{6}$ <112> type, where $b_1 + b_2 + b_3 = 0$. The ITB configuration was the same as those in previous reports [9,30,31]. The atomic-scale structure of the twin-structured samples projected along [1$\bar{1}$0], as shown in Figure 1b. Tensile simulations on these metallic samples were conducted to reveal the deformation mechanisms of these twin-structured nanocrystals. The classical EAM potential [37,39–41] was used for the MD uniaxial tensile simulations. A free boundary condition was applied in the x direction, with set periodic boundary conditions in the other directions. The ambient temperature was maintained at 298 K throughout the simulation process using the Nose–Hoover thermostat. The MD time step was fixed at 1 fs. Before tensile loading, the system was relaxed in an NPT ensemble for 100 ps to obtain the equilibrated structures. The pressure was set to zero in the y-axial direction and there was no pressure control in the other directions. Then, the "fix deform" command of LAMMPS was used to stretch the samples at a constant engineering strain rate of $\dot{\varepsilon} = 1.0 \times 10^8$ s$^{-1}$ along the y-axis in the canonical NVT ensemble. The evolution of the atomic structure during the deformation process was obtained using the analytical method of Ackland and Jones [43]. The atomistic structures were visualized by using Open Visualization Tool (OVITO) software [44].

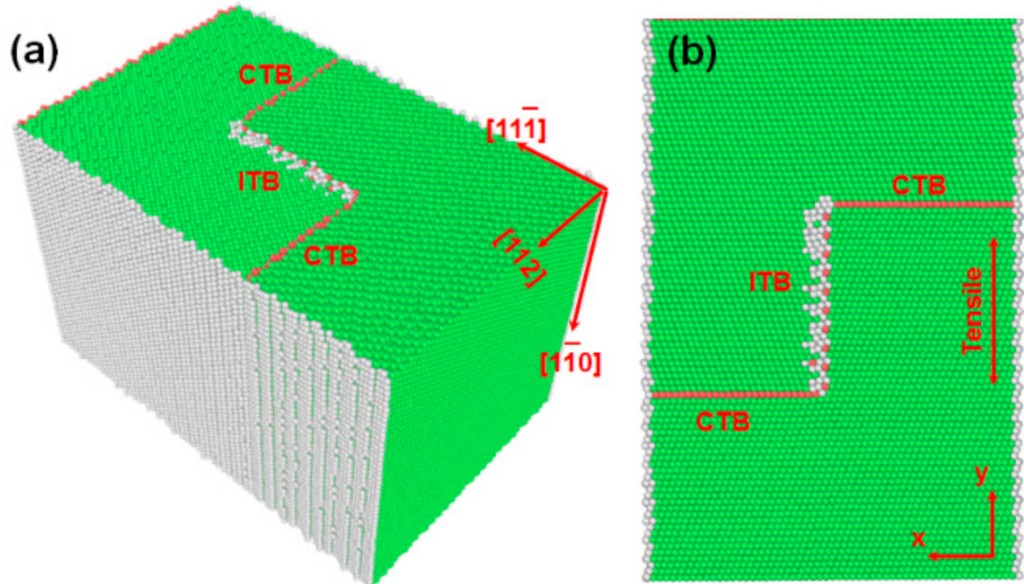

**Figure 1.** Atomistic model of the twin-structured nanocrystal with ITB. (**a**) Three-dimensional structure of the twinned nanocrystals, with the tensile direction along [11$\bar{1}$]; (**b**) a typical image of the twin-structured nanocrystal viewed along [1$\bar{1}$0]. Atom colors correspond to the crystal structure according to common neighbor analysis. The red atoms are CTBs, the green atoms are FCC, and the other atoms are grey.

## 3. Results

Figure 2 shows the true stress–strain curves of the four types of twin-structured nanocrystals with tensile loading. These data were calculated using LAMMPS software and the strain–stress curves were drawn using the ORIGIN software. From these curves, we can see that the strain–stress curves of the nanocrystals increased linearly and then dropped at the yield point. As shown in Figure 2a–d, only the twin-structured Au nanocrystals exhibited strain hardening. As shown in Figure 2a, the yield stress of the twin-structured Au nanocrystal was 3.1 GPa. After yielding, the stress decreased gradually to ~ 2 GPa. With further straining, the stress increased from ~2 to ~ 2.81 GPa. This indicates that there was an obvious strain hardening for the twin-structured Au nanocrystal. As shown in Figure 2b, for the Al nanocrystal, the yield stress was 3.97 GPa. After yielding, the stress dropped sharply to ~2.18 GPa, and there was only a slight increase in the stress with further straining. This indicates that the strain hardening of the twin-structured Al nanocrystal was not obvious. For twin-structured Ni and Cu nanocrystals, as can be seen from Figure 2c,d, the corresponding yield stresses were 11.77 and 6.97 GPa, respectively. After yielding, the stresses dropped sharply, and there was no obvious increase in the stresses with further straining. This indicates that there was no obvious strain hardening for the twin-structured Ni and Cu nanocrystals. According to previous experimental and simulation results [8–12,45,46], the mechanical properties of materials were directly related to their deformation mechanisms, so these differing stress–strain curves indicate that the deformation mechanisms of these twin-structured metals should be different, as discussed below.

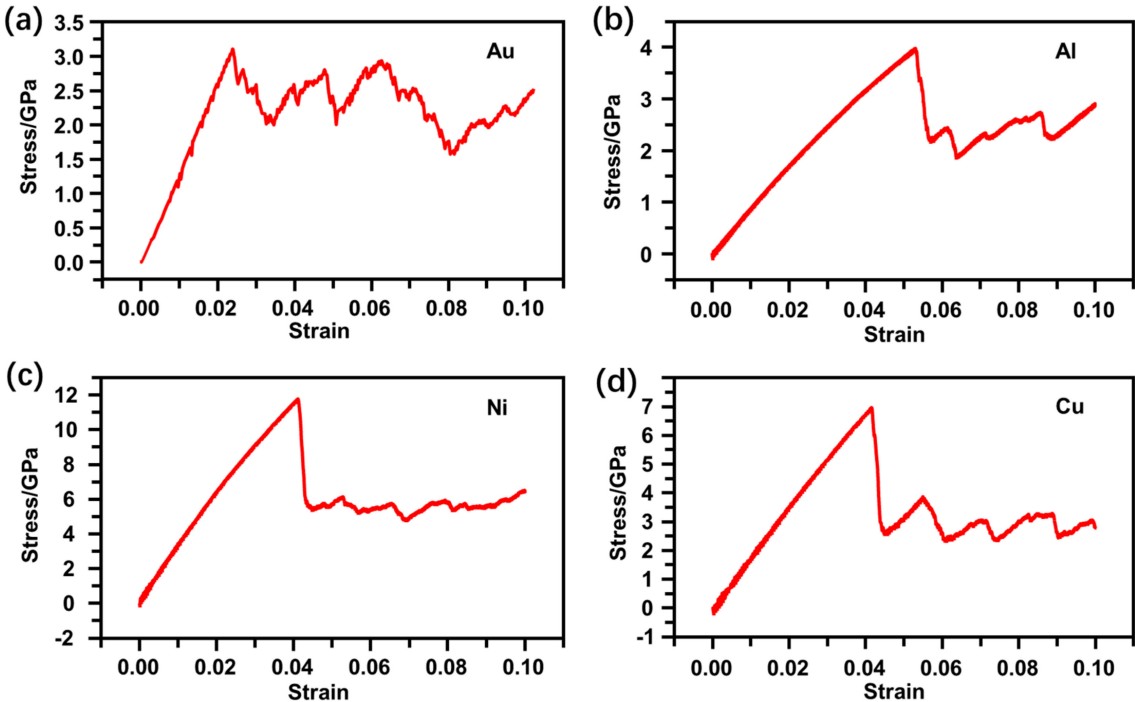

**Figure 2.** Stress–strain curves of (**a**) Au, (**b**) Al, (**c**) Ni, and (**d**) Cu nanocrystals with ITB.

Figure 3 shows the structural evolution process of the Au nanocrystal with ITB during tensile loading. During plastic deformation, both partial dislocations intersecting with the CTB and partial dislocations reacting with each other were observed. Figure 3a shows a typical image of the ITB before loading. This ITB consists of periodical sets of partial dislocations, " $b_1$, $b_2$, $b_3$," which is consistent with previous reports [9,30,31]. With tensile loading, as shown in Figure 3b, two partial dislocations were nucleated from the free surface and glided into the interior of the nanocrystals. This partial dislocation nucleation and gliding resulted in SFs, as shown by the red atoms in the figures. At the same time, the partial dislocations at the ITB also glided into the FCC lattice (left side) on every third (111) plane, leading to periodical sets of short SFs observed in the ITB. This partial dislocation activity led to the original yield of the Au nanocrystal. With further straining, two different strain-hardening mechanisms were observed. As shown in Figure 3c–e, the nucleated partial dislocations intersected the CTB, a strengthening deformation model that leads to strong strain hardening [47–50]. In addition to partial dislocation–CTB intersection, partial dislocation–partial dislocation interactions were also observed. From Figure 3f,g, one can see that two partial dislocations reacted with each other to form an L–C lock (as indicated by the arrow in Figure 3g). With further straining, as can be seen from Figure 3g,h, a new partial dislocation was emitted from the free surface and glided toward the right side of the nanocrystal. This partial dislocation intersected with the previously generated SF to form another L–C lock (as indicated by the arrow in Figure 3g). At the same time, the first L–C lock was destroyed. This L–C lock can also provide strong strain hardening [51–53]. Thus, the observed strain hardening should result from the observed partial dislocation–CTB intersection and from the generation of L–C locks due to partial dislocation interactions.

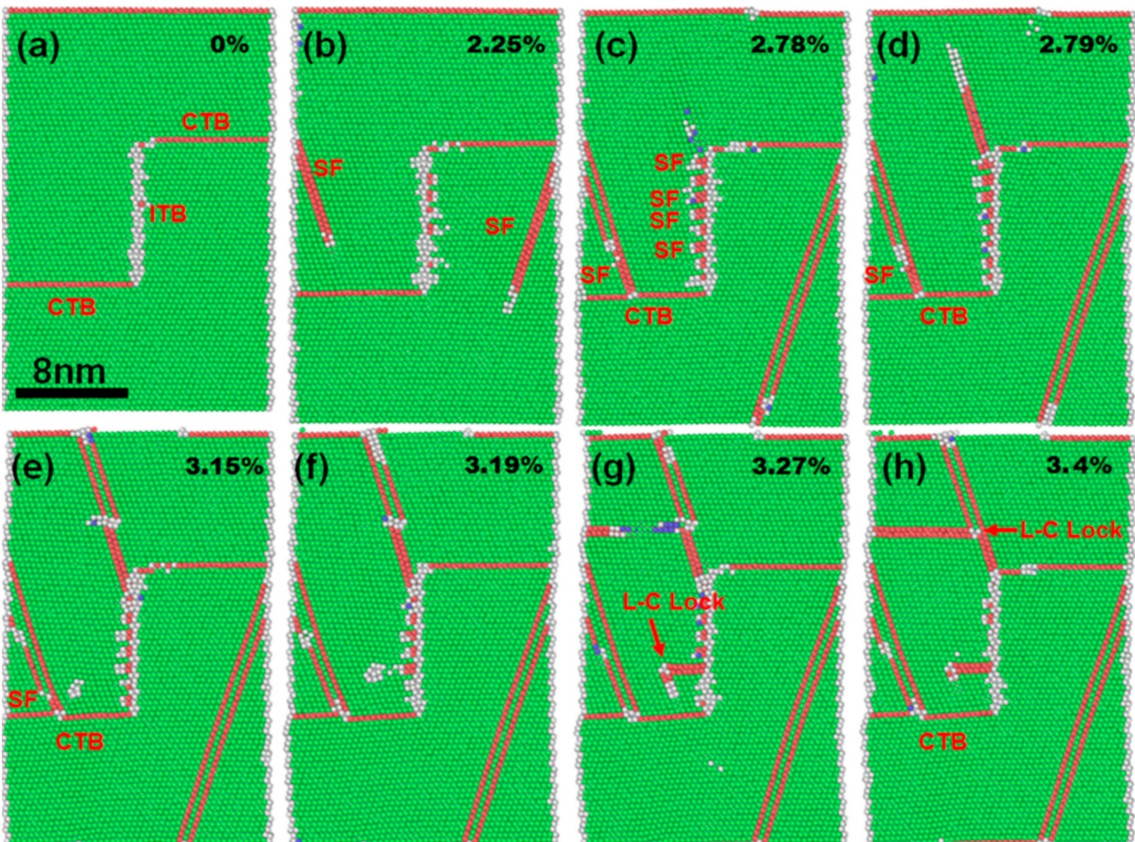

**Figure 3.** Atomic image of a twin-structured Au nanocrystal with tensile strains of 0%, 2.25%, 2.78%, 2.79%, 3.15%, 3.19%, 3.27%, and 3.4%. (**a–d**) Partial dislocations nucleate from the free surface and intersect the CTB; (**e,f**) two partial dislocations react with each other to form a L–C lock; (**g,h**) with further straining, a new partial dislocation intersects with the previous generated SF to form another L–C lock; at the same time, the first L–C lock is destroyed. Red: hexagonal close-packed atoms; green: FCC atoms; gray: other types of atoms.

Compared to Au, the deformation of Al nanocrystals was controlled by reversible ITB migration, while full dislocation and partial dislocation activities were rarely observed. Figure 4 shows a series of images presenting the deformation process of Al nanocrystals with ITB. Figure 4a shows an image of the ITB before loading, where the ITB was straight and the structure was the same as that shown in Figure 3. With loading, as shown in Figure 4b, the thickness of the ITB increased. With continuous loading, as indicated by the dashed red line in Figure 4c–e, the ITB moved toward the left side and the originally straight ITB became irregular. During this migration process, the partial dislocations at the middle of the ITB moved quickly toward the left side, as indicated by the yellow arrow, while those near the CTB moved slowly, as indicated by the black arrows, leading to the originally straight ITB becoming irregular. With further loading, the ITB moved back toward the right side, leading to the irregular ITB becoming relatively straight, as indicated by the dashed yellow line in Figure 4f–j. This process was accomplished via partial dislocations moving back to their initial positions. In addition to ITB migration, full dislocations were also observed in the nanocrystals. As indicated by the red arrows in Figure 4f, two full dislocations were observed. During the deformation process, dislocation–CTB intersection and dislocation interactions were rarely observed. This was consistent with the strain–stress curve shown in Figure 2b, in which no obvious strain hardening occurred in Al with ITB.

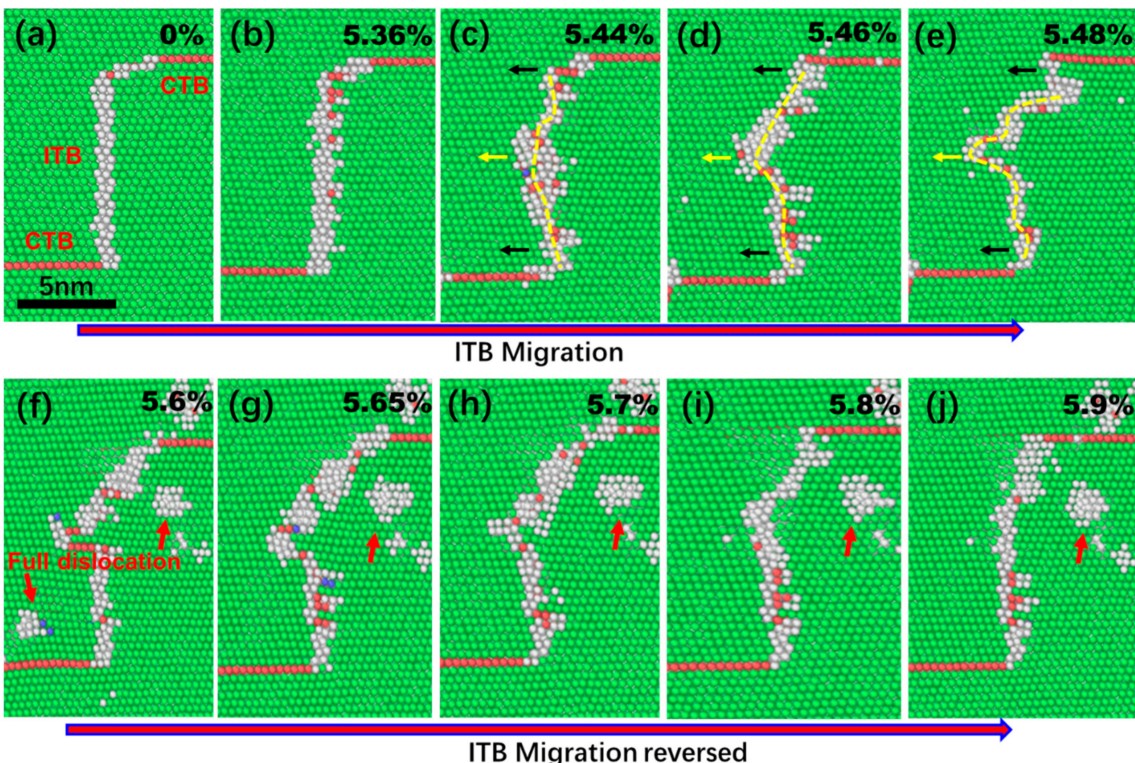

**Figure 4.** The atomic image of a twin-structured Al nanocrystal with tensile strains of 0, 5.36, 5.44, 5.46, 5.48, 5.60, 5.65, 5.70, 5.80, and 5.90%. (**a–e**) The ITB moves toward the left side and the originally straight ITB becomes irregular; (**f–j**) the irregular ITB moves back toward the right side.

For Ni, our results showed that the plastic deformation was controlled by partial dislocations emitted from the ITB or the tips of the SFs, unlike the mechanisms shown in Figures 3 and 4. As shown in Figure 5a–c, a partial dislocation at the ITB glided into the FCC lattice on every third (111) plane, leading to periodic sets of short SFs observed at the ITB. The gliding of this partial dislocation away from its initial position in the ITB led to the yielding of the Ni nanocrystal. As the strain increased, the length of the SFs increased, as these partial dislocations continued to move to the left side of the Ni nanocrystal, as shown in Figure 5d. With continuous straining, the ITB can serve as a dislocation source for partial dislocation emission.

The arrow in Figure 5e indicates that a new partial dislocation emission from the ITB that resulted in a SF (denoted as $SF_1$) was observed. During deformation, this newly nucleated dislocation can also serve as a dislocation source. As shown in Figure 5f, two new partial dislocations that nucleated from the tip of the SFs were observed (denoted as $SF_2$ and $SF_3$). This process resulted in three new SFs observed in the nanocrystals. During further straining, as shown in Figure 5f–h, partial dislocations that nucleated from the tips of the SFs were frequently observed, leading to a high density of SFs (denoted as $SF_1$–$SF_5$) in the nanocrystal.

For the Cu nanocrystal, it seems that the plastic deformation mechanisms were similar to those of the Ni nanocrystals, as shown in Figure 6. Comparing the structure of the ITB shown in Figure 6a–c, a high density of short SFs was observed at the ITB, which resulted from the partial dislocation at the ITB gliding into the FCC lattice, which was similar to those observed in Figure 5a–d. In addition, as indicated by the arrow in Figure 6c, new partial dislocations nucleated from the ITB, resulting in the generation of a SF. As the strain was increased, another partial dislocation was nucleated from the CTB and glided into the nanocrystal, resulting in another SF (Figure 6d,e). With further loading, we could see a new partial dislocation nucleated from the tip of the newly generated SFs (Figure 6e,f), similar to that observed in Ni nanocrystals.

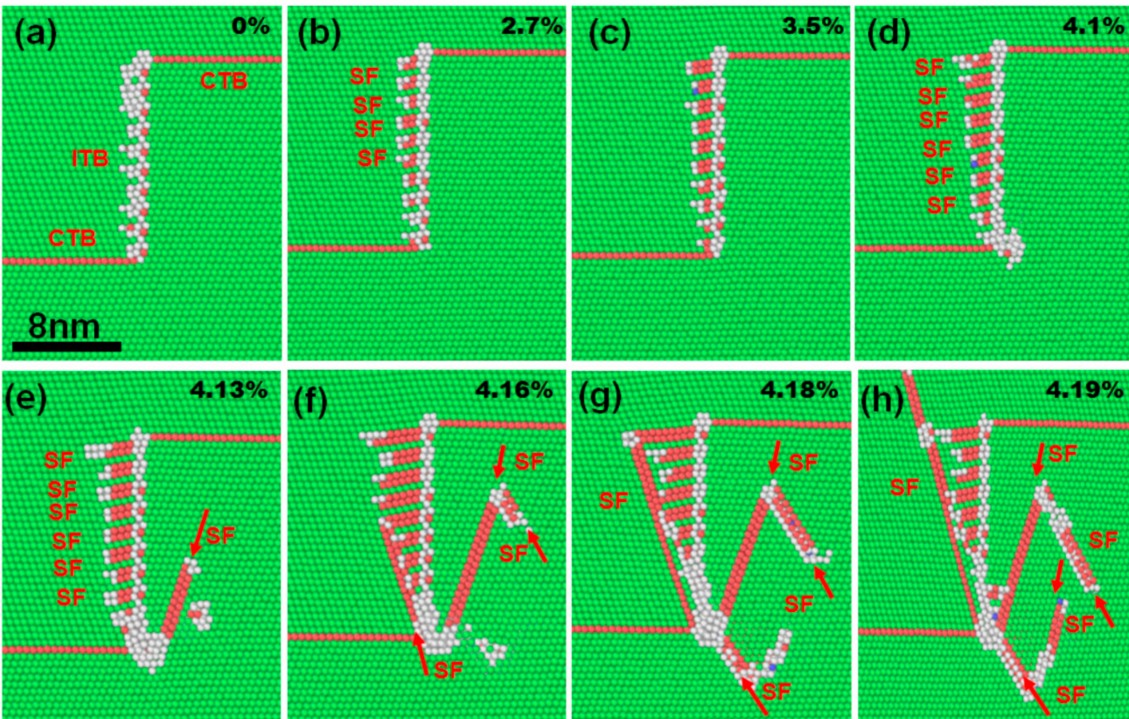

**Figure 5.** Series of images showing the deformation process of twin-structured Ni nanocrystals: (**a**–**d**) the partial dislocation at the ITB glides into the FCC lattice on every third (111) plane, leading to periodical sets of short SFs observed at the ITB; (**e**) the ITB serves as a partial dislocation source; (**f**–**h**) direct observation of partial dislocations emitted from the ITB or the tips of SFs.

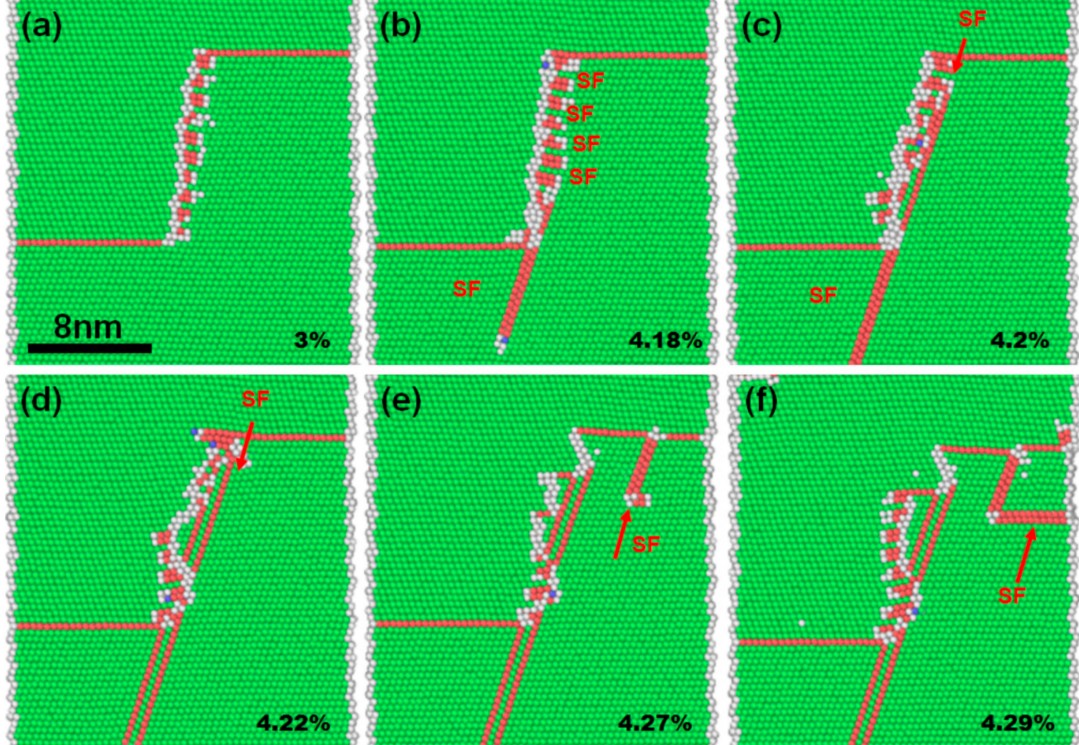

**Figure 6.** Series of images showing the deformation process of twin-structured Cu nanocrystal: (**a**–**d**) the partial dislocations at the ITB glide into the FCC lattice, leading to periodical sets of short SFs observed at the ITB; (**e**) the ITB serves as a partial dislocation source; (**f**) direct observation of partial dislocations emitted from ITB or the tips of SFs.

## 4. Discussion

Here, our results revealed three deformation mechanisms for metals with ITBs. The first deformation mechanism was observed in Au nanocrystals with ITBs, in which the plasticity is governed by partial dislocations nucleated from the free surface of the nanocrystal, intersecting with the CTBs or reacting with each other to form L–C locks. This deformation mechanism could contribute to both strain hardening and high plasticity [25,28,47,53]. The second deformation mechanism was observed in Al nanocrystals with ITBs, in which the plasticity is governed by reversible ITB migration and full dislocations [6,8,24,37]. The third type of deformation was observed in both Ni and Cu nanocrystals with ITBs, in which the plasticity is governed by partial dislocations emitted from the ITB or the tips of the SFs [28–32]. The L–C lock formation has been reported in nanocrystalline metals [51,54], while it has rarely been reported in nanocrystals. This L–C lock can also lead to the formation of dislocation networks and thereby resist the propagation of mobile dislocations [51,54–56]. As reported by Fu et al. [57], once a partial is pinned by the L–C locks, it can hider ITB migration and delay the plasticity process (as shown in Figures 2a and 3), leading to the strain hardening. In addition, Li et al. studied the interaction of dislocations with CTBs in nanocrystalline Cu, which can impede the dislocation motion [35]. This is consistent with Figure 3c–e, which show that the interaction between dislocations and a CTB results in the formation of steps on the CTB. Results similar to the second deformation mechanism have been reported at a crack tip after stress relaxation [29], while our results show that continuous deformation can also lead to reversible ITB migration. Previous studies [31,32] have shown that the ITB always maintains a straight-line shape during the process, unlike the results of the current simulation, as shown in Figure 4c–e. In the third mechanism, numerous partial dislocations emitted from the ITB or the tips of the SFs create a 9R structure. This mechanism is frequently observed in experiments [31,58,59], supporting the validity of our simulation. Previous studies have indicated that ITB migration can also occur through the movement of this 9R structure [58]. While SFs serving as dislocation sources have rarely been reported in experiments, several important studies have also shown that TBs can significantly affect the physical and chemical properties of materials [60,61]. Thus, our results may also provide clues for understanding how and why the physical and chemical properties of materials usually change with different structures and boundaries [62,63], especially during deformation and practical application. Meanwhile, by comparing the simulation results with the experimental results, we can achieve a more comprehensive understanding of the deformation mechanisms of twin-structured nanocrystals.

It is well established that the SF energy ($\gamma_{sf}$) of Ni is higher than those of Au, Al, and Cu [64–66], and while the deformation in Al nanocrystals is governed by reversible ITB migration and full dislocations, the deformation in both Ni and Cu nanocrystals with ITBs is governed by partial dislocations. This indicates that the observed deformation model in nanocrystal metals with ITBs cannot be explained in terms of the absolute value of the $\gamma_{sf}$. We believe such deformation mechanisms can be understood in terms of the ratio $\gamma_{sf}/\gamma_{usf}$ (unstable SF energy), which is 0.97 for Al, 0.55 for Ni, 0.41 for Au, and 0.23 for Cu [64–68]. Indeed, the energy barrier that has to be overcome for creating a trailing partial is very low when $\gamma_{sf}/\gamma_{usf}$ approaches unity [67], allowing the observation of full dislocations in MD simulations. However, when the $\gamma_{sf}/\gamma_{usf}$ is low, such as Ni, Au, or Cu, the energy increase necessary for nucleating the trailing partial is substantial, which suggests that plasticity may occur through partial dislocation activities [67,68]. Interestingly, values of $\gamma_{sf}/\gamma_{usf}$ can also affect the migration of ITBs and the dislocation activities at ITBs. For those metals with relativity low values of $\gamma_{sf}/\gamma_{usf}$, partials are more likely to be emitted from the ITB, creating a 9R structure [58], as in Au (Figure 3c), Ni (Figure 5d), and Cu (Figure 6b), while for metals with high values of $\gamma_{sf}/\gamma_{usf}$, plasticity was mainly governed through ITB migration without partial dislocations.

## 5. Conclusions

In summary, MD simulations were used to investigate the deformation mechanisms of metallic Au, Al, Ni, and Cu nanocrystals with ITBs. We discovered that the deformation mechanisms of nanocrystals with ITBs in these four metals differ, and they include: partial dislocation intersections with CTBs or reactions with each other, reversible ITB migration, and emission of partial dislocations from the ITB or the tips of the SFs. We showed that the dislocation–CTB interaction or reaction with each other could contribute to both strain hardening and plasticity. Our MD simulations combined with previous experimental observations provide a deeper insight into the fundamental deformation mechanisms in metals, and may also provide clues for achieving high strength and high plasticity of twin-structured alloy nanocrystalline metals.

**Author Contributions:** Y.T. and L.W. designed the project and guided the research. Y.T. and L.F. prepared the manuscript. Y.Z. and Z.W. performed the simulation. Y.Z., L.F. and L.W. analyzed the data and prepared the figures. L.F. and L.W. reviewed the manuscript. All authors have read and agreed to the published version of the manuscript.

**Funding:** This work was funded by the Beijing Natural Science Foundation (Grant No. Z180014), and the Beijing Outstanding Young Scientists Projects (Grant No. BJJWZYJH01201910005018).

**Institutional Review Board Statement:** Not applicable.

**Informed Consent Statement:** Not applicable.

**Data Availability Statement:** The data presented in this study are available on request from the corresponding author.

**Conflicts of Interest:** The authors declare no conflict of interest.

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
