# Peer review of "Deformation Mechanisms of FCC-Structured Metallic Nanocrystal with Incoherent Twin Boundary"

_metals, doi:10.3390/met11111672_

Round 1
Reviewer 1 Report
The article presents interesting results but lacks a broader analysis, verification and deeper discussion of the results.
The article presents interesting results of deformation mechanisms of FCC-structured nanocrystals were molecular dynamic (MD) simulations were investigated. However, the study lacks verification of simulations and reference to real experiments and analyzes.
How were the graphs shown in Figure 2 obtained? Please comment on the course of the presented charts.
In the paragraph above Fig. 2, the authors refer to the relationship between mechanical properties and deformation mechanisms presented in the literature. Do the authors have their own results or are they based on the literature?
For example: Figure 3 shows the structural evolution process of the Au nanocrystals with ITB during tensile loading. The presented results apply to the simulation results. Can you propose an experiment here, in which you can verify / compare the obtained results? Similar comments apply to the subsequent simulations.
It must be said that, the all the simulation results presented are very interesting. However, comparing the theoretical results with the experimental results would be very valuable and would give a broader view of the topic presented.
To be able to say that “..Our experimental results provide a deeper insight into the fundamental deformation mechanisms in metals and provide clues for achieving high strength and high plasticity of twin-structured alloy NC metals ..” (in Conclusions), one should refer to the experimental results and demonstrate that simulations give a proper view of the observed phenomena and properties. Especially since there are known experiments that allow the analysis of structural deformations caused e.g. by stresses (Fig. 3). Otherwise, these are only the results of a computer simulation carried out with unclear conditions and assumptions.
Before publishing, I would suggest developing the discussions and results.
Author Response
Response to reviewer
We appreciate the referees’ in-depth reviews. The manuscript has been improved substantially because of these constructive comments. A response to each point raised is included below, where the review comments (in italics) are first repeated and then our responses follow.
Summary: The article presents interesting results but lacks a broader analysis, verification and deeper discussion of the results.
Response: We thank the reviewer for the positive comments. We have revised the manuscript as per the reviewer’s suggestions.
Comment 1: The article presents interesting results of deformation mechanisms of FCC-structured nanocrystals were molecular dynamic (MD) simulations were investigated. However, the study lacks verification of simulations and reference to real experiments and analyzes.
Response: We appreciate the reviewer’s constructive comments, we have added a discussion in the revised manuscript. The details are provided below.
“The L-C lock formation has been reported in nanocrystalline metals [51,54], while it has rarely been reported in nanocrystals. This L-C lock can also lead to formation of dislocation networks and thereby resist the propagation of mobile dislocations [51,54−56]. As reported by Fu et al. [57], once a partial is pinned by the L-C locks, it can hider ITB migration and delay the plasticity process (as shown in Figure 2a and Figure 3), leading to the strain hardening. In addition, Li et al. have studied the interaction of dislocations with CTBs in nanocrystalline Cu, which can impede the dislocation motion [35]. This is consistent with Figures 3c-e, which show that the interaction between dislocations and a CTB results in the formation of steps on the CTB. Results similar to the second deformation mechanism have been reported at a crack tip after stress relaxation [29], while our results show that continuous deformation can also lead to reversible ITB migration. Previous studies [31,32] show that the ITB always maintains a straight line shape during the process, unlike the results of the current simulation as shown in Figures 4c-e. In the third mechanism, numerous partial dislocations emitted from the ITB or the tips of the SFs create a 9R structure. This mechanism is frequently observed in in experiments [31,58,59], supporting the validity of our simulation. Previous studies have indicated that ITB migration can also occur through the movement of this 9R structure [58].”.
Comment 2: How were the graphs shown in Figure 2 obtained? Please comment on the course of the presented charts.
Response: Following the reviewer's comment, we have added several sentences in the revised manuscript. The details are provided below:
“These data were calculated using LAMMPS software and the strain-stress curves were drawn using the ORIGIN software. From these curves, we can see that the strain-stress curves of the nanocrystals increase linearly and then drop at the yield point”.
Comment 3: In the paragraph above Fig. 2, the authors refer to the relationship between mechanical properties and deformation mechanisms presented in the literature. Do the authors have their own results or are they based on the literature?
Response: The relationship between mechanical properties and deformation mechanisms presented in manuscript are based on a large number of previous studies and our experiments. In order to better respond to the reviewers' comment, we have added several sentences to the revised manuscript. The details are provided below.
“According previous experimental and simulation results [8−12,45,46], the mechanical properties of materials are directly related to their deformation mechanisms, so these differing stress-strain curves indicate that the deformation mechanisms of these twin-structured metals should be different, as will be discussed below (Figure 3–Figure 6).”.
Comment 4: For example: Figure 3 shows the structural evolution process of the Au nanocrystals with ITB during tensile loading. The presented results apply to the simulation results. Can you propose an experiment here, in which you can verify/compare the obtained results? Similar comments apply to the subsequent simulations.
Response: Following the reviewer's constructive suggestion, we have compared our simulation results with previous experimental studies. We notice that some of the experimental results are consistent with our simulation, while our simulation also provides some different findings that have not been reported in experiments. This is mainly due to the fact that the camera speed is unable to capture the process, and there is a lack of in-situ experimental evidences, especially atomic-scale in situ experimental evidences. In response to the reviewer's suggestion, we have added a discussion to the revised manuscript, provided below:
“The L-C lock formation has been reported in nanocrystalline metals [51,54], while it has rarely been reported in nanocrystals. This L-C lock can also lead to formation of dislocation networks and thereby resist the propagation of mobile dislocations [51,54−56]. As reported by Fu et al. [57], once a partial is pinned by the L-C locks, it can hider ITB migration and delay the plasticity process (as shown in Figure 2a and Figure 3), leading to the strain hardening. In addition, Li et al. have studied the interaction of dislocations with CTBs in nanocrystalline Cu, which can impede the dislocation motion [35]. This is consistent with Figures 3c-e, which show that the interaction between dislocations and a CTB results in the formation of steps on the CTB.”.
Comment 5: It must be said that, the all the simulation results presented are very interesting. However, comparing the theoretical results with the experimental results would be very valuable and would give a broader view of the topic presented.
Response: Following the reviewer's constructive suggestion, we have comparing our results with previous experimental results. Several sentences were added in the revised manuscript. The details are provided below:
“The L-C lock formation has been reported in nanocrystalline metals [51,54], while it has rarely been reported in nanocrystals. This L-C lock can also lead to formation of dislocation networks and thereby resist the propagation of mobile dislocations [51,54−56]. As reported by Fu et al. [57], once a partial is pinned by the L-C locks, it can hider ITB migration and delay the plasticity process (as shown in Figure 2a and Figure 3), leading to the strain hardening. In addition, Li et al. have studied the interaction of dislocations with CTBs in nanocrystalline Cu, which can impede the dislocation motion [35]. This is consistent with Figures 3c-e, which show that the interaction between dislocations and a CTB results in the formation of steps on the CTB. Results similar to the second deformation mechanism have been reported at a crack tip after stress relaxation [29], while our results show that continuous deformation can also lead to reversible ITB migration. Previous studies [31,32] show that the ITB always maintains a straight line shape during the process, unlike the results of the current simulation as shown in Figures 4c-e. In the third mechanism, numerous partial dislocations emitted from the ITB or the tips of the SFs create a 9R structure. This mechanism is frequently observed in in experiments [31,58,59], supporting the validity of our simulation. Previous studies have indicated that ITB migration can also occur through the movement of this 9R structure [58]. While SFs serving as dislocation sources have rarely been reported in experiments, several important studies also show that TBs can significantly affect the physical and chemical properties of materials [60,61]. Thus, our results may also provide clues for understanding how and why the physical and chemical properties of materials usually changed with different structures and boundaries [62,63], especially during deformation and practical application. Meanwhile, by comparing the simulation results with the experimental results, we can achieve a more comprehensive understanding the deformation mechanisms of twin-structured nanocrystals.”.
Comment 6: To be able to say that “.Our experimental results provide a deeper insight into the fundamental deformation mechanisms in metals and provide clues for achieving high strength and high plasticity of twin-structured alloy NC metals.” (in Conclusions), one should refer to the experimental results and demonstrate that simulations give a proper view of the observed phenomena and properties. Especially since there are known experiments that allow the analysis of structural deformations caused e.g. by stresses (Fig. 3). Otherwise, these are only the results of a computer simulation carried out with unclear conditions and assumptions.
Response: Following the reviewer's constructive suggestion, we have added a discussion comparing our simulation results with previous experimental results. In addition, we have also revised the corresponding sentence as below:
“Our MD simulations combined with previous experimental observations provide a deeper insight into the fundamental deformation mechanisms in metals, and also may provide clues for achieving high strength and high plasticity of twin-structured alloy nanocrystalline metals.”.

Reviewer 2 Report
The manuscript entitled "Deformation Mechanism of FCC-Structured Metallic Nanocrystal with Incoherent Twin Boundary" by Y. Tao et. al. presents simulation work on the deformation mechanism in the case of 4 metallic nanostructures. Although I consider the topic to be of interest to "Metals" journal readership, the the English style and grammar in some paragraphs makes the manuscript hard to follow. Moreover a complete justification of the experiment is missing in my opinion. I find that a further review stage is needed prior to publication. In the following I present the aspects which require improvement:
- The English should be thoroughly checked. I present here only a few examples: lines 38-40 "While in real twin-structured metals, there is also a high density of incoherent twin boundaries (ITBs), which can significantly affect their mechanical properties.", lines 56-58 "The twin-structured Au, Al, Ni and Cu nanocrystals with two CTB and an ITB, which was formed from the contacted of the two inverted “L” shape parts, and then relaxed. These twin-structured nanocrystals with a square cross-section.", lines 65-68 "The sample with the length (x-axis along the [112] direction), width (y-axis along the [111̅] direction), and thickness (z-axis along the [11̅0] direction) of 12 nm, 18 nm, and 12 nm, respectively. The atomic-scale structure of the twin-structure samples projected along the [11̅0] as shown in Figure 1b." The authors are encouraged not to limit their English check only to these examples.
- The authors should provide an extended justification of their experiment. The rationale behind the choice for metals, potential application areas, current shortcomings in the field.
- Even though the proposed manuscript presents experimental data, the authors should include in the Discussion paragraph possible envisioned experimental methods which can demonstrate the simulation results. For example the author should consider scanning probe techniques (e.g., atomic force microscopy: 2020. Impact of stacking faults and domain boundaries on the electronic transport in cubic silicon carbide probed by conductive atomic force microscopy. Advanced Electronic Materials, 6(2), p.1901171.) and also optical techniques (second harmonic generation microscopy: 2014. Nonlinear optical imaging of defects in cubic silicon carbide epilayers. Scientific reports, 4(1), pp.1-6).
- The text formatting error in line 31 should be corrected.
Author Response
Response to reviewer
We appreciate the referees’ in-depth reviews. The manuscript has been improved substantially because of these constructive comments. A response to each point raised is included below, where the review comments (in italics) are first repeated and then our responses follow.
Summary: The manuscript entitled "Deformation Mechanism of FCC-Structured Metallic Nanocrystal with Incoherent Twin Boundary" by Y. Tao et. al. presents simulation work on the deformation mechanism in the case of 4 metallic nanostructures. Although I consider the topic to be of interest to "Metals" journal readership, the English style and grammar in some paragraphs makes the manuscript hard to follow. Moreover a complete justification of the experiment is missing in my opinion. I find that a further review stage is needed prior to publication. In the following I present the aspects which require improvement.
Response: We thank the reviewer for the positive comments. We have revised the manuscript as per the reviewer’s suggestions.
Comment 1: The English should be thoroughly checked. I present here only a few examples: lines 38-40 "While in real twin-structured metals, there is also a high density of incoherent twin boundaries (ITBs), which can significantly affect their mechanical properties.", lines 56-58 "The twin-structured Au, Al, Ni and Cu nanocrystals with two CTB and an ITB, which was formed from the contacted of the two inverted “L” shape parts, and then relaxed. These twin-structured nanocrystals with a square cross-section.", lines 65-68 "The sample with the length (x-axis along the [112] direction), width (y-axis along the [111̅] direction), and thickness (z-axis along the [11̅0] direction) of 12 nm, 18 nm, and 12 nm, respectively. The atomic-scale structure of the twin-structure samples projected along the [11̅0] as shown in Figure 1b." The authors are encouraged not to limit their English check only to these examples.
Response: We thank the reviewer for this comment. We have asked a native speaker to polish our manuscript thoroughly.
Comment 2: The authors should provide an extended justification of their experiment. The rationale behind the choice for metals, potential application areas, current shortcomings in the field.
Response: Following the reviewer's constructive suggestion, we have added a discussion in the revised manuscript. The details are provided below.
“The L-C lock formation has been reported in nanocrystalline metals [51,54], while it has rarely been reported in nanocrystals. This L-C lock can also lead to formation of dislocation networks and thereby resist the propagation of mobile dislocations [51,54−56]. As reported by Fu et al. [57], once a partial is pinned by the L-C locks, it can hider ITB migration and delay the plasticity process (as shown in Figure 2a and Figure 3), leading to the strain hardening. In addition, Li et al. have studied the interaction of dislocations with CTBs in nanocrystalline Cu, which can impede the dislocation motion [35]. This is consistent with Figures 3c-e, which show that the interaction between dislocations and a CTB results in the formation of steps on the CTB. Results similar to the second deformation mechanism have been reported at a crack tip after stress relaxation [29], while our results show that continuous deformation can also lead to reversible ITB migration. Previous studies [31,32] show that the ITB always maintains a straight line shape during the process, unlike the results of the current simulation as shown in Figures 4c-e. In the third mechanism, numerous partial dislocations emitted from the ITB or the tips of the SFs create a 9R structure. This mechanism is frequently observed in in experiments [31,58,59], supporting the validity of our simulation. Previous studies have indicated that ITB migration can also occur through the movement of this 9R structure [58].”.
Comment 3: Even though the proposed manuscript presents experimental data, the authors should include in the Discussion paragraph possible envisioned experimental methods which can demonstrate the simulation results. For example the author should consider scanning probe techniques (e.g., atomic force microscopy: 2020. Impact of stacking faults and domain boundaries on the electronic transport in cubic silicon carbide probed by conductive atomic force microscopy. Advanced Electronic Materials, 6(2), p.1901171.) and also optical techniques (second harmonic generation microscopy: 2014. Nonlinear optical imaging of defects in cubic silicon carbide epilayers. Scientific reports, 4(1), pp.1-6).
Response: We thank the reviewer for providing us these two interesting papers, and we have added several sentences to discuss the research technology that can be considered in the future. The details are given below:
“While SFs serving as dislocation sources have rarely been reported in experiments, several important studies also show that TBs can significantly affect the physical and chemical properties of materials [60,61]. Thus, our results may also provide clues for understanding how and why the physical and chemical properties of materials usually changed with different structures and boundaries [62,63], especially during deformation and practical application. Meanwhile, by comparing the simulation results with the experimental results, we can achieve a more comprehensive understanding the deformation mechanisms of twin-structured nanocrystals.”.
Comment 4: The text formatting error in line 31 should be corrected.
Response: We would like to thank the reviewer for the suggestion. We have revised the formatting error accordingly.

Reviewer 3 Report
Interesting paper. Major revisions are in order for the authors to address the comments detailed below:
Language needs to be revised by a native speaker. Several errors found in the text. Please revise carefully.
“Because the mechanical properties of materials are directly related to their deformation mechanism [6−11],”: see also other recent works such as 10.1016/j.scriptamat.2021.114219 and 10.1016/j.matdes.2020.108505 where this is evidenced and further complement the introduction.
“While in real twin-structured metals”: why real? Unclear.
“was formed from the contacted of the two inverted “L” shape parts, and then relaxed”: why this selection? Clarify.
More details on the simulations are required. Please add.
Stress/strain curves: why don’t the curves start at 0 stress for fig 2 a? This is weird. Also are these curves experimental or simulated?
“some of the partial dislocations at the”: can this be quantified?
The simulation is rather nice, but is there any experimental data (even from literature) to compare these results with? This would be very interesting to provide.
What was the dimensions of the nanoscrystals imposed?
Author Response
Response to reviewer
We appreciate the referees’ in-depth reviews. The manuscript has been improved substantially because of these constructive comments. A response to each point raised is included below, where the review comments (in italics) are first repeated and then our responses follow.
Summary: Interesting paper. Major revisions are in order for the authors to address the comments detailed below.
Response: We thank the reviewer’s comments.
Comment 1: Language needs to be revised by a native speaker. Several errors found in the text. Please revise carefully.
Response: We thank the reviewer for this comment. We have asked a native speaker to polish our manuscript thoroughly.
Comment 2: “Because the mechanical properties of materials are directly related to their deformation mechanism [6−11],”: see also other recent works such as 10.1016/j.scriptamat.2021.114219 and 10.1016/j.matdes.2020.108505 where this is evidenced and further complement the introduction.
Response: We thank the reviewer for providing us these two interesting papers. We have cited the papers in the revised manuscript.
Comment 3: “While in real twin-structured metals”: why real? Unclear.
Response: We note the word “real” may mislead the reader. Therefore, we have deleted the word and revised the manuscript accordingly.
Comment 4: “was formed from the contacted of the two inverted “L” shape parts, and then relaxed”: why this selection? Clarify.
Response: Following the reviewer's comment, we have provided details for explanation in this part. The details are provided below:
“We adopt a rectangular model with periodic boundary condition along the x-axis. The model is divided into a two-part computational cell, which was formed from the combination of two “L” shaped parts in order to construct a twin-structured nanocrystal, as shown in Figure 1. Then, this model was relaxed ensuring the structure with minimum energy. The nanocrystal contains a Σ3 {112} ITB with the thickness of 30 (111) planes, and two CTBs separated by the ITB, as shown in Figure 1.”.
Comment 5: More details on the simulations are required. Please add.
Response: Following the reviewer's suggestion, we have provide more details in this part. The details are provided below:
“We adopt a rectangular model with periodic boundary condition along the x-axis. The model is divided into a two-part computational cell, which was formed from the combination of two “L” shaped parts in order to construct a twin-structured nanocrystal, as shown in Figure 1. Then, this model was relaxed ensuring the structure with minimum energy. The nanocrystal contains a Σ3 {112} ITB with the thickness of 30 (111) planes, and two CTBs separated by the ITB, as shown in Figure 1. The deformations of twin-structured Au, Al, Ni, and Cu nanocrystals were conducted using MD simulations at room temperature, ~298K. These twin-structured nanocrystals with a square cross-section contain about 24,000 atoms. The coordinate systems are the x-axis along [112], the y-axis along [11-1], and the z-axis along [1-10]. The dimensions of the nanocrystals are 12 nm for both the x- and z-axes, and 18 nm for the y-axis, respectively. In the three-dimensional structure of the sample as shown in Figure 1a, the ITB (gray atoms) was parallel to the longitudinal direction of the nanocrystals, and the CTBs (red atoms) were perpendicular to that direction. The stretch simulations were conducted by using a large-scale atomic/molecular massively parallel simulator (LAMMPS) program [38], with the classical embedded-atom method (EAM) potential [39−42]. The ITB consists of periodical sets of partials dislocation “b1, b2, b3” with b=1/6<112> type, where b1+b2+b3=0. The ITB configuration was the same as that in previous reports [9,30,31]. The atomic-scale structure of the twin-structure samples projected along [1-10] as shown in Figure 1b. Tensile simulations on these metallic samples were conducted to reveal the deformation mechanisms of these twin-structure nanocrystals. The classical EAM potential [37,39−41] was used for the MD uniaxial tensile simulations. A free boundary condition was applied in the x direction, with set periodic boundary conditions in the other directions. The ambient temperature was maintained at 298 K throughout the simulation process using the Nose–Hoover thermostat. The MD time step was fixed at 1 fs. Before tensile loading, the system was relaxed in an NPT ensemble for 100 ps to obtain the equilibrated structures. The pressure was set to zero in the y-axial direction and there was no pressure control in other directions. Then the ‘fix deform’ command of LAMMPS was used to stretch the samples at a constant engineering strain rate of ε'=1.0×108 s-1 along the y-axis in the canonical NVT ensemble. The evolution of the atomic structure during the deformation process was obtained using the analytical method of Ackland and Jones [43]. The atomistic structures were visualized by using Open Visualization Tool (OVITO) software [44].”.
Comment 6: Stress/strain curves: why don’t the curves start at 0 stress for fig 2 a? This is weird. Also are these curves experimental or simulated?
Response: We apologize for this mistake in our presentation. The strain and stress data in Figure 2 was calculated by the LAMMPS software and the curves were drawn by the ORIGIN software. Figure 2a accidentally ignores the data near zero strain in the process of drawing. In fact, the stress at zero strain should be zero. Figure 2a has been corrected accordingly.
Comment 7: “some of the partial dislocations at the”: can this be quantified?
Response: during the deformation, the motion speeds of these dislocations are different from each other, and this leads to the ITB being curved. It is difficult to quantify these differing speeds. In order to describe this process clearly, we have added different colored arrows in the Figures to indicate qualitatively the speeds and revised the manuscript accordingly.
“During this migration process, the partial dislocations at the middle of the ITB move quickly toward the left side, as indicated by the yellow arrow, while those near the CTB move slowly, as indicated by the black arrows, leading to the originally straight ITB becoming irregular.”.
Comment 8: The simulation is rather nice, but is there any experimental data (even from literature) to compare these results with? This would be very interesting to provide.
Response: Following the reviewer's constructive suggestion, we have added a discussion to the revised manuscript. The details are provided below:
“The L-C lock formation has been reported in nanocrystalline metals [51,54], while it has rarely been reported in nanocrystals. This L-C lock can also lead to formation of dislocation networks and thereby resist the propagation of mobile dislocations [51,54−56]. As reported by Fu et al. [57], once a partial is pinned by the L-C locks, it can hider ITB migration and delay the plasticity process (as shown in Figure 2a and Figure 3), leading to the strain hardening. In addition, Li et al. have studied the interaction of dislocations with CTBs in nanocrystalline Cu, which can impede the dislocation motion [35]. This is consistent with Figures 3c-e, which show that the interaction between dislocations and a CTB results in the formation of steps on the CTB. Results similar to the second deformation mechanism have been reported at a crack tip after stress relaxation [29], while our results show that continuous deformation can also lead to reversible ITB migration. Previous studies [31,32] show that the ITB always maintains a straight line shape during the process, unlike the results of the current simulation as shown in Figures 4c-e. In the third mechanism, numerous partial dislocations emitted from the ITB or the tips of the SFs create a 9R structure. This mechanism is frequently observed in in experiments [31,58,59], supporting the validity of our simulation. Previous studies have indicated that ITB migration can also occur through the movement of this 9R structure [58]. While SFs serving as dislocation sources have rarely been reported in experiments, several important studies also show that TBs can significantly affect the physical and chemical properties of materials [60,61]. Thus, our results may also provide clues for understanding how and why the physical and chemical properties of materials usually changed with different structures and boundaries [62,63], especially during deformation and practical application. Meanwhile, by comparing the simulation results with the experimental results, we can achieve a more comprehensive understanding the deformation mechanisms of twin-structured nanocrystals.”.
Comment 9: What was the dimensions of the nanocrystals imposed?
Response: Following the reviewer's comments, we have revised the manuscript accordingly. The details are provided below.
“The coordinate systems are the x-axis along [112], the y-axis along [11-1], and the z-axis along [1-10]. The dimensions of the nanocrystals are 12 nm for both the x- and z-axes, and 18 nm for the y-axis, respectively.”.

Round 2
Reviewer 2 Report
An improved version of the manuscript was submitted by the authors. I consider it ready for publication in the present form.
Reviewer 3 Report
Very good paper. All comments were addressed and acceptance is recommended.